# Forewarned Is Forearmed: Machine Learning Algorithms for the Prediction of Catheter-Induced Coronary and Aortic Injuries

**DOI:** 10.3390/ijerph192417002

**Published:** 2022-12-18

**Authors:** Jacek Klaudel, Barbara Klaudel, Michał Glaza, Wojciech Trenkner, Paweł Derejko, Marek Szołkiewicz

**Affiliations:** 1Department of Invasive Cardiology and Interventional Radiology, St. Adalbert’s Hospital, Copernicus PL, 80-462 Gdańsk, Poland; 2Department of Cardiology, St. Vincent de Paul Hospital, Pomeranian Hospitals, 81-348 Gdynia, Poland; 3Department of Decision Systems and Robotics, Faculty of Electronics, Telecommunications and Informatics, Gdansk University of Technology, 80-233 Gdańsk, Poland; 4Department of Cardiology, Medicover Hospital, 02-972 Warszawa, Poland; 5Cardiac Arrhythmias Department, National Institute of Cardiology, 04-628 Warszawa, Poland; 6Department of Cardiology and Interventional Angiology, Kashubian Center for Heart and Vascular Diseases, Pomeranian Hospitals, 84-200 Wejherowo, Poland

**Keywords:** catheter-induced dissection, coronary artery dissection, aortocoronary dissection, iatrogenic complications, dissection predictors, machine-learning

## Abstract

Catheter-induced dissections (CID) of coronary arteries and/or the aorta are among the most dangerous complications of percutaneous coronary procedures, yet the data on their risk factors are anecdotal. Logistic regression and five more advanced machine learning techniques were applied to determine the most significant predictors of dissection. Model performance comparison and feature importance ranking were evaluated. We identified 124 cases of CID in electronic databases containing 84,223 records of diagnostic and interventional coronary procedures from the years 2000–2022. Based on the f1-score, Extreme Gradient Boosting (XGBoost) was found to have the optimal balance between positive predictive value (precision) and sensitivity (recall). As by the XGBoost, the strongest predictors were the use of a guiding catheter (angioplasty), small/stenotic ostium, radial access, hypertension, acute myocardial infarction, prior angioplasty, female gender, chronic renal failure, atypical coronary origin, and chronic obstructive pulmonary disease. Risk prediction can be bolstered with machine learning algorithms and provide valuable clinical decision support. Based on the proposed model, a profile of ‘a perfect dissection candidate’ can be defined. In patients with ‘a clustering’ of dissection predictors, a less aggressive catheter and/or modification of the access site should be considered.

## 1. Introduction

Catheter-induced coronary dissections are iatrogenic complications of low incidence (0.09–0.14%) but have a serious impact on inpatient morbidity and mortality due to their ostial-proximal location (i.e., jeopardising a large myocardial area) and the potential for a retrograde progression to the aorta (reported incidence of 0.02%) [1,2,3]. Along with coronary perforation and stroke, they are among the most dreaded complications of percutaneous coronary procedures, in which the inpatient mortality can reach 6% for coronary injuries and 22% for dissections propagating into the aorta [1,3]. Previous analyses of coronary artery dissections included various device-related triggers, such as guidewire insertion, balloon inflation, or stent deployment, with catheter-induced injuries typically accounting for the minority of cases [4]. The few studies that exclusively included catheter-induced dissections analysed mainly their prevalence, management and clinical outcomes [1,2]. In the studies, detailed characteristics of the ‘non-dissected population’ were not available and therefore predictive modelling was not possible. The data on the risk factors of catheter-induced dissection have, thus far, been limited to case reports, case series and reviews, and have not been validated in large cohorts of patients [5,6].

The prediction of cardiovascular adverse events has been traditionally based on logistic regression modelling, itself a machine learning technique, with other, more advanced machine-learning algorithms gaining popularity only recently [7,8]. Several studies comparing conventional risk assessment methods with machine learning models in patients undergoing percutaneous coronary angioplasty reported a significantly improved performance and discrimination of the latter, while others showed only a modest improvement [8,9,10,11].

Based on a population of over 80,000 catheterised patients, we sought to establish clinical, anatomical and procedural predictors of catheter-induced dissection. Defining them can help create a profile of a high-risk patient where special vigilance and precautions are required. It can also help identify the injury faster and prevent its serious propagation.

## 2. Materials and Methods

### 2.1. Study Population and Data Collection

We performed a retrospective study of iatrogenic dissections of coronary arteries, aorta and coronary bypass grafts induced by a diagnostic, guiding or guide extension catheter, based on the electronic databases of two high-volume centres with a similar caseload and catchment area, serving a joint population of 2305 million. During the whole study period, overall, 20 operators and cardiology fellows worked in the catheterisation laboratories.

The dissection cases were retrieved from dedicated registries of procedural complications prospectively collected at both departments. A double check was performed using a database query with keywords to identify missing cases. Coronary dissections caused by a stent deployment, rotational atherectomy, balloon inflation or rupture, wire manipulation, and other non-catheter-related mechanisms were excluded. All relevant cineangiograms were adjudicated by two experienced interventionists, with the controversies reviewed by both. In the case of a disagreement, a third reviewer was included to reach a consensus.

### 2.2. Definitions

Atypical coronary artery take-off definition included: an anomalous origin from the opposite sinus of Valsalva, ectopic high coronary artery take-off (at least 10 mm above the sinotubular junction), anteriorly displaced right coronary artery (originating from the anterior one third of the right sinus), and anomalous origin of the left coronary artery from the pulmonary artery (Bland–White–Garland syndrome). ‘Stenotic ostium’ was defined as atheromatous plaque with a ≥30% lumen narrowing, while ‘small ostium’ meant a vessel origin of ≤3 mm in diameter. National Heart, Lung, and Blood Institute (NHLBI) and Dunning classifications were used for coronary and aortocoronary dissections, respectively [3]. As per the NHLBI classification, type A dissection represents the minor radiolucent areas (luminal haziness) within the lumen with no persistence of contrast; type B are linear dissections, i.e., parallel tracts or a double lumen separated by a radiolucent area, with minimal or no persistence of contrast; type C dissections appear as contrast outside the coronary lumen (“extraluminal cap”) with the persistence of contrast; type D are spiral luminal filling defects, usually with excessive contrast staining of the false lumen (“barbershop pole”); type E dissections appear as persistent lumen defects with a delayed antegrade flow, while type F dissections represent a total occlusion of the coronary lumen. Dunning et al. proposed a classification of iatrogenic aortic dissection into three grades: Type 1 is defined as a focal dissection restricted to the coronary cusp; Type 2 involves the sinus of Valsalva and extends above the sinotubular junction up the ascending aorta but is less than 4 cm; and in Type 3, dissection extends beyond the aortic sinuses up the ascending aorta greater than 4 cm [3]. 

### 2.3. Statistical Analysis

Multivariable logistic regression modelling and five other machine-learning algorithms (Decision Tree, Random Forest, Naive Bayes, K-Nearest Neighbors, and Extreme Gradient Boosting—XGBoost) were applied to determine the independent predictors of a catheter-induced dissection. The candidate variables were preselected from clinically relevant pre-catheterisation characteristics, procedural factors and angiographic findings. A covariance matrix of the candidate variables was evaluated, with highly correlated variables removed (Appendix A). A univariate analysis was then used to assess which of the potential candidate variables had a statistical association with dissection (*p* < 0.05) with the results obtained using the Fisher exact test (Appendix A). A significance level of *p* < 0.05 was used throughout the analysis. The odds ratios were reported with a 95% confidence interval.

The values were evaluated using a three-fold cross validation due to the small number of dissections in the validation dataset. Data were randomly split into a training (70%) and a testing set (30%). The training dataset was oversampled using the Synthetic Minority Oversampling Technique (SMOTE) to account for the class imbalance. A model performance comparison was performed with the assessment of accuracy, precision, recall, and f1-score. Precision is a measure of how many dissections detected by the algorithm were true dissections. Recall measures how many dissections of all true dissections were correctly detected by the model. The f1-score shows a harmonic mean between the precision and recall. The feature importance ranking was evaluated for each model. Sklearn 1.1.3, xgboost 1.7.1, imbalanced-learn 0.9.1, and dalex 1.5.0 libraries for Python 3.9.5 were used for the modelling.

The study was approved by the regional research ethics committee. As the study used data from a retrospective registry, the requirement for informed consent was waived. All patient information was anonymised and de-identified before analysis. All the procedures performed in this study involving human participants were in accordance with the Declaration of Helsinki (as revised in 2013).

## 3. Results

### 3.1. Dissection Incidence

Overall, 84,223 diagnostic and therapeutic coronary procedures were performed between June 2000 and August 2022 in the two centres, and 124 cases of catheter-induced dissection were identified (Graphical Abstract). There were 115 coronary dissections (including 12 aortocoronary injuries) plus six cases of trauma limited to the sinus of Valsalva (including two injuries extending above the sinotubular junction) and two saphenous vein graft injuries (Table 1). The total incidence of catheter-induced dissection was 0.147%, with the aortic involvement in 0.023% (19 cases). Given that 84% of dissections had been inflicted by a guiding catheter (including two guide extension catheter-related dissections), the dissection rate for angioplasty procedures was higher and reached 0.196% (Table 2). As for diagnostic angiography, the overall dissection rate was three times lower at 0.064%. The left and right coronary artery were almost equally affected, in 57 and 58 cases, respectively (Table 1). One of the major left main coronary artery branches was selectively intubated and thus injured in 12 cases, with a short left main coronary artery (<15 mm) found in 9 of them (75%). Localised dissection as per the NHLBI classification (Types A and B) accounted for 45% of the coronary and graft dissections, while the more serious Types C to F accounted for 55%, including 9.4% of the acute occlusions (Type F) (Table 1).

### 3.2. Clinical and Procedural Characteristics of the Dissection Cohort

The mean age of the dissection cohort was 69.1 (±11.9) years (Table 2). Both genders were almost equally represented (with 47.6% females vs. 52.4% males). The most prevalent feature in the medical history of the patients was arterial hypertension, prior percutaneous coronary intervention (PCI), prior myocardial infarction and diabetes mellitus. Most patients presented with acute coronary syndrome (73%), including 41% admitted with the diagnosis of STE-ACS (ST-elevation acute coronary syndrome). The in-hospital mortality was 5.6% (seven patients), with three deaths attributed to the dissection or its direct consequences. Radial access was utilised in 80% of the dissection cases. Most dissections occurred during coronary intervention as opposed to diagnostic angiography, in 84 vs. 16%, respectively. 

### 3.3. Logistic Regression Modelling

In the logistic regression analysis, the use of a guiding catheter—PCI (odds ratio [OR], 7.49; 95% confidence interval [CI], 4.72–11.87), small and/or stenotic coronary ostium (OR, 5.53; 95% CI, 3.88–7.88), atypical origin of a coronary artery (OR, 4.99; 95% CI, 1.83–13.6), arterial hypertension (OR, 4.98; 95% CI, 1.25–7.15), and cardiogenic shock (OR, 4.59; 95% CI, 2.0–10.47) were the five strongest predictors of dissection, followed by peripheral arterial disease, chronic obstructive pulmonary disease, chronic renal failure, acute myocardial infarction, and radial access (Figure 1, Appendix A).

### 3.4. Multivariable Logistic Regression vs. Other Machine Learning Methods

The results of the logistic regression modelling were corroborated by the other machine learning techniques. A guiding catheter (PCI) and small/stenotic ostium were the strongest or the second strongest predictor in all but one machine learning algorithm (Figure 2). Similarly, hypertension and radial access scored high in the feature importance ranking, being among the five strongest dissection risk factors in all six techniques.

### 3.5. Model Performance Comparison

All the risk prediction techniques achieved a high or at least reasonable precision (Table 3). Based on the f1-score metric, XGBoost was found to have the optimal balance between a positive predictive value (precision) and sensitivity (recall). As by the best algorithm, namely, XGBoost, the strongest dissection predictors were angioplasty—the use of a guiding catheter, small/stenotic ostium, radial access, female gender, hypertension, atypical coronary origin, chronic total occlusion procedure, acute myocardial infarction, peripheral arterial disease, and prior angioplasty (Figure 2).

## 4. Discussion

### 4.1. Traditional Risk Prediction Modelling

Predictors of catheter-induced coronary and/or aortic dissections have not been systematically studied so far, most likely due to the low incidence of the complication and the consequent paucity of data. However, while no studies have specifically addressed the risk factors of catheter-induced injuries, it is reassuring to observe that several of the dissection predictors in our analysis were the same as those used in popular risk scores for the prediction of PCI in-hospital adverse events, including comorbidities such as hypertension, chronic renal failure, chronic obstructive pulmonary disease, or peripheral arterial disease, as well as the diagnosis of acute myocardial infarction, cardiogenic shock, and prehospital cardiac arrest. Older studies predominantly used logistic regression modelling, while more recent analyses employed other machine learning-based algorithms as well. They typically utilised two types of collected data, i.e., pre-catheterisation variables (e.g., demographic and clinical parameters) and angiographic factors (e.g., coronary anatomy and lesion characteristics). Many of them do not seem to be adverse event-specific as they were used as variables for the prognostication of bleeding, acute kidney injury and/or mortality. 

In a simplified risk score based on the National Cardiovascular Data Registry (NCDR) from the United States, cardiogenic shock, renal function, and age were found to be the most predictive of in-hospital mortality after PCI, with angiographic variables providing only modest incremental information to preprocedural risk assessments [7]. Among other variables included were the PCI status (ST-elevation myocardial infarction vs. no ST-elevation infarction), peripheral vascular disease and chronic lung disease. A study by Brennan et al. found that cardiogenic shock and procedure urgency were the most powerful predictors of inpatient mortality, whereas the presence of CTO was among the most significant angiographic predictors in the high-risk PCI subset [12]. The latest iteration of the risk score based on the NCDR, namely, the CathPCI Registry, reported that procedural urgency, cardiovascular instability, and the level of consciousness after cardiac arrest were the most predictive of in-hospital mortality in the pre-catheterisation model, de-emphasising the importance of coronary anatomy or procedural factors [13]. Two risk stratification models for in-hospital death in patients undergoing PCI based on the Japanese-PCI (J-PCI) registry were developed, namely, a full and a pre-catheterisation one [14]. The pre-catheterisation model included age, sex, clinical presentation, previous PCI, previous coronary artery bypass grafting, hypertension, renal dysfunction/dialysis, peripheral vascular disease, chronic lung disease, cardiopulmonary arrest on arrival, and cardiogenic shock within 24 h, while the full model added angiographic information. Another analysis developed and validated a risk score for the prediction of 30 day mortality after PCI that did not require knowledge of the coronary anatomy [15]. The five variables making up the risk score were cardiogenic shock, advanced age, procedural urgency, history of renal disease and diabetes mellitus. 

In our study, several of the above-mentioned risk factors of PCI complications have been found to be predictors of catheter-induced coronary and/or aortic dissection, specifically hypertension, chronic renal failure, peripheral arterial disease, chronic obstructive pulmonary disease, cardiogenic shock, cardiac arrest, previous PCI, and diabetes mellitus. Cardiogenic shock is one of the strongest predictors of a poor outcome and adverse events [15]. Shock at admission and prehospital cardiac arrest are both markers of a patient’s poor condition and together with an acute myocardial infarction diagnosis they can act as a surrogate for a procedure’s urgency, also associated with adverse outcomes in the above-mentioned studies.

### 4.2. Discriminator: Risk by the Machines

All of the previously described models were developed with traditional logistic regression modelling, while some new risk scores proposed recently have included other, more advanced machine learning algorithms for risk prediction. A model developed for the prediction of in-hospital mortality after PCI compared a logistic regression with three advanced machine-learning algorithms (i.e., AdaBoost, XGBoost and Random Forest) in a population of 479,804 patients and 2549 death events [9]. The authors found that one of the machine-learning algorithms (specifically AdaBoost) exhibited the highest discriminatory performance. In another study, XGBoost and logistic regression were compared in predicting the risk of myocardial infarction in a population-based cohort of over 500,000 subjects, i.e., the UK Biobank [16]. The algorithm was trained on 3013 patients and tested on 7998 patients with a suspected myocardial infarction. Both the regression and XGBoost were equally precise, with the regression model classifying more of the healthy persons correctly and XGBoost being more accurate in identifying individuals who later suffered a myocardial infarction; however, the XGBoost outperformed the logistic regression model according to the receiver operator characteristic scoring better in terms of accuracy in this metric. In contrast, Niimi et al. compared NCDR-CathPCI risk scores for adverse periprocedural events (acute kidney injury, bleeding, and in-hospital mortality) with XGBoost models using data from a prospective, all-comer, multicentre registry from Japan, containing the records of 24,848 patients [8]. The XGBoost model modestly improved the discrimination for acute kidney injury and bleeding but not for in-hospital mortality, while it overestimated the risk for in-hospital mortality in low-risk patients. The authors concluded that the improvement in the overall risk prediction with the machine-learning-based technique was minimal.

Recall (sensitivity) was the most significant metric for us since failing to identify a dissection-prone patient (a Type II error, i.e., the misclassification of a high-risk patient to a low-risk class) has more severe consequences than a false positive classification of a low-risk patient as being at a high risk of a catheter-induced injury (a Type I error). Additionally, while a higher recall comes at the cost of lower precision, precision itself still has to have a reasonable value. To balance a high recall and high precision, the f1-score is used, which shows a harmonic mean between the two. We have compared the performance of all the algorithms and the XGBoost produced the best results in terms of the f1-score, having the highest recall score of all the models with a reasonable precision. This is in line with the current practices of data science, where XGBoost is often treated as the default strategy for tabular datasets [10,16]. Overall, in our study, the tree-based models (i.e., XGBoost, Decision Tree, and Random Forest) performed the best, outperforming the other machine learning algorithms, as also reported by other authors [11].

### 4.3. Anatomical Determinants of Dissection

The only anatomical factors assessed in our model were an atypical coronary origin, chronic total occlusion, and small and/or stenotic coronary ostium. Other angiographic features such as the culprit lesion type, bifurcation disease or vessel tortuosity, all indisputably increase the difficulty of a procedure and, thus, the risk of dissection; however, they are not harvestable in our databases, and clinical factors such as myocardial infarction or cardiogenic shock can be considered surrogates for the procedure’s difficulty. Moreover, many of the models described above were based exclusively on preprocedural factors or developed two types of risk scores, namely, a pre-catheterisation and a full one [7,13,15].

The variant anatomy of the coronary ostia or anomalous origin of the coronary artery is often associated with more difficult coronary intubation due to the lack of a dedicated catheter curve and the resulting misalignment of the catheter and its inferior stability, which may facilitate dissection. Similarly, a small diameter of the coronary ostium and/or significant ostial disease may cause a catheter tip to hit and/or wedge in the vessel wall leading to its dissection. 

A recent study reported a high, 1.8% incidence of aortocoronary dissection during angioplasty of chronic total occlusion in their series [17]. The recanalisation of chronic coronary occlusion is definitely one of the most demanding and complex coronary interventions, carrying a high risk not only of perforation but a catheter-induced dissection as well; however, this risk was typically reported in populations with a high rate of retrograde and subintimal tracking techniques [18,19,20].

### 4.4. Procedural Risk Factors

Our model evaluated several procedural parameters: the use of a guiding catheter (PCI vs. diagnostic angiography), radial vs. femoral access, and right vs. left coronary artery intubation. Several studies have suggested that the use of a guiding catheter (and thus a PCI procedure) is a strong risk factor for iatrogenic coronary and/or aortic dissection [1,2,3,5,19,20,21,22]. Compared to diagnostic angiography, PCI typically involves not only larger French size and more aggressive catheters giving extra support, but also more extensive catheter manipulation with frequent repositioning and deeper insertion for active backup, multiple contrast injections, and a longer indwelling time. The strong correlation between dissection and PCI persists despite twice the number of catheters and coronary intubations performed during diagnostic angiography.

Prior PCI may be a marker of more advanced coronary artery disease (not readily visible as a discrete stenosis) and also of minor injuries sustained during previous catheterisations. Some studies have suggested that cumulative, repeated catheter penetration of the left main coronary artery during consecutive PCI procedures may lead to accelerated stenosis of the ostial-proximal segment of the vessel [23].

In our analysis, the right coronary artery was injured in almost as many cases as the left (57 vs. 58, respectively), and was not found to be a significant dissection predictor. However, in keeping with the findings of Lopez-Minguez et al., aortic dissection started in the right coronary artery and/or involved the right sinus of Valsalva in more than twice as many patients than in the left coronary artery and/or sinus (13 vs. 6 cases, respectively) [24]. As described by Lopez-Minguez et al., in contrast to the left coronary artery, the periostial wall and sinotubular junction of the right coronary artery are formed by less smooth muscle cells and by a less dense matrix of collagen type-I fibres.

To date, two other studies have observed a higher rate of radial access in patients with a catheter-induced dissection, although data on the overall access rates in the general population have not been available [1,2]. This is likely the consequence of more difficult catheter manipulation and poorer catheter stability associated with the technique.

### 4.5. Management

Catheter-induced dissection management has been outlined in several publications [5,25]. Stent deployment is currently the most preferred treatment option due to its immediacy and the low risk of restenosis, especially in segments without significant plaque. Several unique techniques have been proposed for the treatment of severe, especially spiral coronary dissections, such as a cutting balloon used for intramural hematoma decompression or its aspiration with a microcatheter, retrograde recanalisation of the dissected vessel, or an IVUS-guided double guide ping-pong technique [26,27,28]. In the case of injuries involving the aorta, the main treatment strategies include an ostial stent deployment to seal the dissection entry site, watchful waiting with repeat CT angiography, and surgical intervention. As regards the prevention of dissection, several intraprocedural precautions, besides the obligatory watching for pressure damping and coaxial alignment, can be employed. They include the use of a less aggressive curve and/or a smaller size catheter (e.g., a 5 French for PCI or 4 F for angiography), guide extension catheters, quick vessel wiring for guide stabilisation, or a keep-out-wire left in the aortic sinus to avoid deep engagement during a balloon or jailed wire retrieval. 

## 5. Study Limitations

This was a retrospective study with all the limitations inherent in its nature; however, in the case of rare complications, multicentre registries and retrospective studies of large databases collected over long periods of time were necessarily utilised. Our results are based on many years of experience of a high-volume, yet from only two centres. Due to the fast dissemination and standardisation of percutaneous tools and techniques, our model can probably be extrapolated into other populations, taking into account differences in the population mix and prevalence of comorbidities as well as age and gender differences. We could not adjust for several other anatomical and procedural factors that could further refine the risk prediction, such as an operator’s experience, the unfavourable anatomy of access arteries, culprit lesion characteristics, coronary artery calcifications, or coronary disease complexity, as the data on their prevalence in the general population are not systematically collected in databases. We studied some of these factors in the dissection cohort in our previous analysis; however, a comparison with the non-complicated population could not be made for the reasons mentioned above [29]. Another limitation of the study is that it was not externally validated on a separate population. It is essential to assess the generalisability and the reproducibility of a prediction model by using data from other centres; therefore, external validation is warranted.

## 6. Conclusions

Based on several readily ascertainable clinical, procedural and angiographic features, patients at a high risk of catheter-induced dissection can be identified. Risk prediction can be bolstered with machine-learning algorithms that go beyond conventional logistic regression modelling. For patients meeting the criteria of ‘an ideal dissection candidate’, a high level of vigilance as well as extraordinary precautions should be exercised.

## Figures and Tables

**Figure 1 ijerph-19-17002-f001:**
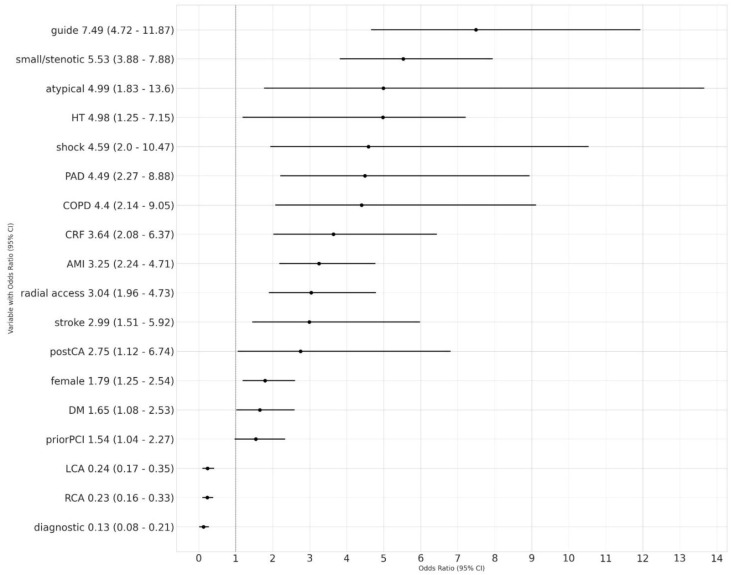
Multivariable logistic regression modelling of catheter-induced dissection predictors. Adjusted OR (point estimate) and 95% CI (error bars) indicate the likelihood ratio of dissection from the logistic regression. OR > 1 indicates increased odds of dissection. CI: confidence interval; OR: odds ratio. AMI: acute myocardial infarction; atypical: atypical origin of coronary artery; CA: cardiac arrest; CI: confidence interval; COPD: chronic obstructive pulmonary disease; CRF: chronic renal failure; DM: diabetes mellitus; guide: use of a guiding catheter (angioplasty); HT: arterial hypertension; LCA: left coronary artery intubation; OR: odds ratio; PAD: peripheral arterial disease; PCI: percutaneous coronary intervention; small/stenotic: coronary ostium ≤3 mm and/or with ≥30% stenosis; RCA: right coronary artery intubation.

**Figure 2 ijerph-19-17002-f002:**
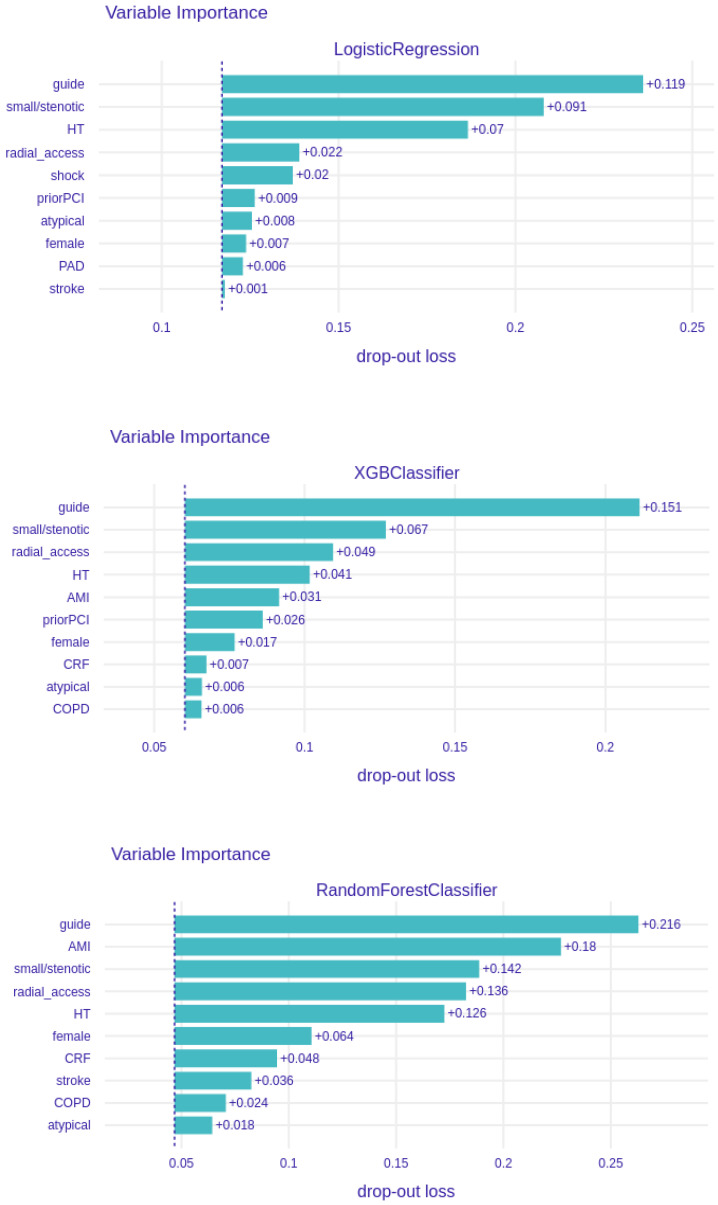
Feature importance ranking. The figure lists the relative importance of variables in all six models used for the prediction of catheter-induced dissections. AMI: acute myocardial infarction; atypical: atypical origin of coronary artery; CA: cardiac arrest; COPD: chronic obstructive pulmonary disease; CRF: chronic renal failure; guide: use of a guiding catheter (angioplasty); HT: arterial hypertension; knn: K-Nearest Neighbors; PAD: peripheral arterial disease; PCI: percutaneous coronary intervention; small/stenotic: coronary ostium ≤3 mm and/or with ≥30% stenosis; xgboost: Extreme Gradient Boosting.

**Table 1 ijerph-19-17002-t001:** Dissection characteristics.

Dissected Vessel; *n* = 117			
LCA	58		49.6
RCA	57		48.7
SVG	2		1.7
Coronary Artery (+SVG) Dissection by NHLBI Type; *n* = 117			
Localised:	A-B	53	45.3
Extensive:	C-F	53	45.3
	C	27	23.1
	D	21	17.9
	E	5	4.3
	F	11	9.4
Aortic Involvement; *n* = 19			
Isolated sinus of Valsalva dissection	7		36.8
Aortocoronary dissection	12		63.2
Aortic dissection starting in RCA and/or involving right SoV	13		68.4
Aortic dissection starting in LCA and/or involving left SoV	6		31.6
Aortic Dissection by Dunning Classification; *n* = 19			
Type I	10		52.6
Type II	5		26.3
Type III	4		21.1
Treatment; *n* = 124			
Conservative	39		31.5
Intervention	83		66.9
Surgery	2		1.6

LCA: the left coronary artery; NHLBI: National Heart, Lung, and Blood Institute; RCA: right coronary artery; SoV: the sinus of Valsalva; SVG: saphenous vein graft.

**Table 2 ijerph-19-17002-t002:** Clinical and procedural characteristics of the dissection cohort.

	*n* = 124	
Age	69.1	11.9
Female gender	59	47.6
Body mass index	27.4	4.5
Left ventricular ejection fraction	48.9	11.8
**Comorbidities and Medical History**		
Diabetes mellitus	27	21.8
Hypertension	75	60.5
Chronic renal failure	14	11.3
Peripheral arterial disease	9	7.3
Stroke	9	7.3
Chronic obstructive pulmonary disease	7	5.6
Prior myocardial infarction	28	22.6
Prior PCI	37	29.8
Prior CABG	6	4.8
**Initial Presentation and in-Hospital Mortality**		
Non-ACS	33	26.6
NSTE-ACS	40	32.3
STE-ACS	51	41.1
Prehospital cardiac arrest	5	4
Shock at admission	6	4.8
In-hospital death	7	5.6
In-hospital death due to dissection	3	2.4
**Procedural characteristics**		
Femoral access	25	20.2
Radial access (incl. ulnar and brachial access)	99	79.8
Coronary angiography	20	16.1
PCI	104	83.9
CTO angioplasty	9	7.3

Values are numbers (%) or means ± standard deviation. ACS: acute coronary syndrome; CABG: coronary artery bypass grafting; CTO: chronic total occlusion; NSTE: non-ST elevation; PCI: percutaneous coronary intervention; STE: ST-elevation.

**Table 3 ijerph-19-17002-t003:** Model Performance Comparison.

	Algorithm	Accuracy	Precision	Recall	f1-Score
**1**	logistic_regression	0.956286	0.043805	0.539876	0.079809
**2**	decision_tree	0.997499	0.690994	0.330623	0.445231
**3**	random_forest	0.997524	0.725000	0.322687	0.443662
**4**	naive_bayes	0.958517	0.050157	0.451607	0.085814
**5**	knn	0.996715	0.429293	0.168990	0.235908
**6**	xgboost	0.997695	0.747911	0.362563	0.488273

Comparison of performance of the six machine-learning models in terms of the accuracy, precision, recall and the harmonic mean of the last two, f1-score (the highest possible f1-score is 1, which means a perfect precision and recall, while the lowest is 0). Knn: K-Nearest Neighbors; xgboost: Extreme Gradient Boosting.

## Data Availability

Data are available from the authors upon reasonable request.

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
