# Peer review of "Forewarned Is Forearmed: Machine Learning Algorithms for the Prediction of Catheter-Induced Coronary and Aortic Injuries"

_ijerph, 2022, doi:10.3390/ijerph192417002_

Round 1
Reviewer 1 Report
In the reviewed paper the authors examined the usefulness of 5 machine learning algorithms in the prediction of dissection of coronary arteries by coronary catheters. Over 84,000 treatment data from 22 years in two centers were analyzed. Overall, the number of identified cases of catheter dissection was 0.023% during coronary angiography and almost 10 times higher during PCI. Data was randomly splited 70/30% between training/testing. Analysis showed the superiority of machine learning algorithms compared to multivariate logistic regression modeling. XGBoost showed the best positive predictive value and sensitivity rates in predicting coronary dissection. The results of the analysis can be considered effective for some patients and therefore preventive actions can be possible.
One small note. The study underestimated one, in my opinion important, factor that is the experience and skills of the operator. It is primarily about the so-called „learning curve” for each of the young operators (it can be assumed that there were such operators in the analyzed group over 22 years) together with the volume of annual procedures. It in no way diminishes the analysis, but should be considered in limitations of the study.
Author Response
We would like to thank the reviewer for his insightful remarks which enabled us to correct the manuscript.
We do acknowledge the fact that both the individual operator’s expertise and centre’s experience may influence the incidence of catheter-induced dissections. We assessed both factors in the dissection subgroup in our previous work (Klaudel J, Glaza M, Klaudel B, Trenkner W, Pawłowski K, Szołkiewicz M. Catheter-induced coronary artery and aortic dissections. A study of the mechanisms, risk factors, and propagation causes. Cardiol J. 2022 Jun 28. doi: 10.5603/CJ.a2022.0050.) Unfortunately, in our databases, operators’ experience is not recorded for all catheterised patients so it could not be used as a candidate variable for initial predictor selection. Our previous analysis, however, has shown that most dissections in our cohort were caused by experienced operators performing urgent interventions in high-risk patients.
We have introduced due corrections in the limitations section:
“We could not adjust for several other anatomical and procedural factors that could further refine risk prediction, such as operator’s experience, unfavourable anatomy of access arteries, culprit lesion characteristics, coronary artery calcifications, or coronary disease complexity, as the data on their prevalence in the general population are not systematically collected in databases. We studied some of these factors in the dissection cohort in our previous analysis; however, a comparison with the non-complicated population could not be made for the reasons mentioned above [29].”
Reviewer 2 Report
Dear authors,
I have studied with interest the manuscript «Forewarned Is Forearmed: Machine-Learning Algorithms for Prediction of Catheter-induced Coronary and Aortic Injuries». The manuscript is well written and the work presented is original. The authors made a very important conclusion that patients at high risk of catheter-induced dissection can be identified. Risk prediction can be bolstered with machine learning algorithms that go beyond conventional logistic regression modelling. For patients meeting the criteria of ‘an ideal dissection candidate’, a high level of vigilance as well as extraordinary precautions should be exercised. All the cited references are relevant to the research and well-balanced. The results are clearly presented and all the conclusions are supported by the results.
I have some comments that could improve the quality of the paper.
1. How have you excluded operator-depended cases?
2. What about coronary artery calcification? Have you analyzed this predictor?
3. But in general, I think that this is a very worthy work. I express my gratitude to the authors for their work and my great pleasure to read their results.
But in general, I think that this is a very worthy work. I express my gratitude to the authors for their work and my great pleasure to read their results.
Author Response
We would like to thank the reviewer for his insightful remarks. As aptly noted by the reviewer, both operator’s experience and factors such as heavy calcification may influence the incidence of catheter-induced dissections. However, both parameters are not recorded in our electronic databases for the general catheterised population so they could not be used in our analysis. We have analysed the operator’s experience in the dissection subgroup in our previous study (Klaudel J, Glaza M, Klaudel B, Trenkner W, Pawłowski K, Szołkiewicz M. Catheter-induced coronary artery and aortic dissections. A study of the mechanisms, risk factors, and propagation causes. Cardiol J. 2022 Jun 28. doi: 10.5603/CJ.a2022.0050.).
An appropriate remark has been added to the limitations section to improve the quality of the paper:
“We could not adjust for several other anatomical and procedural factors that could further refine risk prediction, such as operator’s experience, unfavourable anatomy of access arteries, culprit lesion characteristics, coronary artery calcifications, or coronary disease complexity, as the data on their prevalence in the general population are not systematically collected in databases. We studied some of these factors in the dissection cohort in our previous analysis; however, a comparison with the non-complicated population could not be made for the reasons mentioned above [29].”